# Adaptive Graph Convolutional Recurrent Network for Traffic Forecasting

**Lei Bai**
UNSW, Sydney
baisanshi@gmail.com

**Lina Yao**
UNSW, Sydney
lina.yao@unsw.edu.au

**Can Li**
UNSW, Sydney
can.li4@student.unsw.edu.au

**Xianzhi Wang**
University of Technology Sydney
xianzhi.wang@uts.edu.au

**Can Wang**
Griffith University
can.wang@griffith.edu.au

## Abstract

Modeling complex spatial and temporal correlations in the correlated time series data is indispensable for understanding the traffic dynamics and predicting the future status of an evolving traffic system. Recent works focus on designing complicated graph neural network architectures to capture shared patterns with the help of pre-defined graphs. In this paper, we argue that learning node-specific patterns is essential for traffic forecasting while the pre-defined graph is avoidable. To this end, we propose two adaptive modules for enhancing Graph Convolutional Network (GCN) with new capabilities: 1) a Node Adaptive Parameter Learning (NAPL) module to capture node-specific patterns; 2) a Data Adaptive Graph Generation (DAGG) module to infer the inter-dependencies among different traffic series automatically. We further propose an Adaptive Graph Convolutional Recurrent Network (AGCRN) to capture fine-grained spatial and temporal correlations in traffic series automatically based on the two modules and recurrent networks. Our experiments [1] on two real-world traffic datasets show AGCRN outperforms state-of-the-art by a significant margin without pre-defined graphs about spatial connections.

## 1   Introduction

The fast urbanization introduces growing populations in cities and presents significant mobility and sustainability challenges. Among those challenges, Intelligent Transportation Systems (ITS) has become an active research area [1], given its potential to promote system efficiency and decision-making. As an essential step towards the ITS, traffic forecasting aims at predicting the future status (e.g., traffic flow and speed, and passenger demand) of urban traffic systems. It plays a vital role in traffic scheduling and management and has attracted tremendous attention from the machine learning research community in recent years [2, 3, 4, 5, 6].

Traffic forecasting is challenging due to the complex intra-dependencies (i.e., temporal correlations within one traffic series) and inter-dependencies (i.e., spatial correlations among multitudinous correlated traffic series) [3] generated from different sources, e.g., different loop detectors/intersections for traffic flow & traffic speed prediction, and various stations/regions for passenger demand prediction. Traditional methods simply deploy time series models, e.g., Auto-Regressive Integrated Moving Average (ARIMA) and Vector Auto-Regression (VAR), for traffic forecasting. They cannot capture the nonlinear correlations nor intricate spatial-temporal patterns among large scale traffic

data. Recently, researchers shift to deep-learning-based methods and focus on designing new neural network architectures to capture prominent spatial-temporal patterns shared by all traffic series. They typically model temporal dependencies with recurrent neural networks [7, 8, 9, 10] (e.g., Long-Short Term Memory and Gated Recurrent Unit) or temporal convolution modules [3, 4]. Regarding spatial correlations, they commonly use GCN-based methods [2, 4, 3, 6, 11, 5, 12] to model unstructured traffic series and their inter-dependencies.

While recent deep-learning-based methods achieve promising results, they are biased to the prominent and shared patterns among all traffic series—the shared parameter space makes current methods inferior in capturing fine-grained data-source specific patterns accurately. In fact, traffic series exhibit diversified patterns (as shown in Fig. 1), they may appear similar, dissimilar, and even contradictory owning to the distinct attributes across a variety of data sources [7, 13]. Moreover, existing GCN-based methods require pre-defining an inter-connection graph by similarity or distance measures [14] to capture the spatial correlations. That further requires substantial domain knowledge and is sensitive to the graph quality. The graphs generated in this manner are normally intuitive, incomplete, and not directly specific to the prediction tasks; they may contain biases and not adaptable to domains without appropriate knowledge.

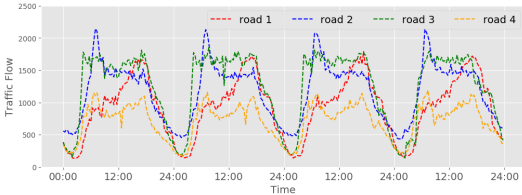

Figure 1: Examples of traffic flow with diverse patterns. The traffic flow of road 3 is steady in the day time. As a contrast, the traffic flows of road 1, 2 and 4 have obvious evening peak, morning peak, and both peaks, respectively.

Instead of designing more complicated network architectures, we propose two concise yet effective mechanisms by revising the basic building block of current methods (i.e., GCN) to solve the above problems separately. Specifically, we propose to enhance GCN with two adaptive modules for traffic forecasting tasks: 1) a Node Adaptive Parameter Learning (NAPL) module to learn node-specific patterns for each traffic series—NAPL factorizes the parameters in traditional GCN and generates node-specific parameters from a weights pool and bias pool shared by all nodes according to the node embedding; 2) a Data Adaptive Graph Generation (DAGG) module to infer the node embedding (attributes) from data and to generate the graph during training. NAPL and DAGG are independent and can be adapted to existing GCN-based traffic forecasting models both separately and jointly. All the parameters in the modules can be easily learned in an end-to-end manner. Furthermore, we combine NAPL and DAGG with recurrent networks and propose a unified traffic forecasting model - Adaptive Graph Convolutional Recurrent Network (AGCRN). AGCRN can capture fine-grained node-specific spatial and temporal correlations in the traffic series and unify the nodes embeddings in the revised GCNs with the embedding in DAGG. As such, training AGCRN can result in a meaningful node representation vector for each traffic series source (e.g., roads for traffic speed/flow, stations/regions for passenger demand). The learned node representation contains valuable information about the road/region and can be potentially applied to other tasks [15].

We evaluate AGCRN on two real-world datasets for the multi-step traffic prediction task and compare it with several representative traffic forecasting models. The experimental results show that AGCRN outperforms state-of-the-art with a significant margin. We also conduct ablation studies and demonstrate the effectiveness of both NAPL and DAGG.

## 2 Related Work

**Correlated time series prediction** Traffic forecasting belongs to correlated time series analysis (or multivariate time series analysis) and has been studied for decades. In recent years, deep learning has dominated the correlated time series prediction due to its superior ability in modeling complex functions and learning correlations from data automatically. A majority of such studies [16, 17, 10, 8, 18, 19, 20, 21] rely on LSTM or GRU to model the temporal dynamics in the time series data. Some efforts employ temporal convolutional networks [22, 23, 24] to enable the model process very long sequence with fewer time. However, these studies do not explicitly model the inter-dependencies among different time series. A very recent work [25] uses transformers for

correlated time series prediction. Such work normally requires massive training samples due to tremendous trainable parameters [26].

**GCN based Traffic forecasting** Different with general correlated time series prediction, traffic forecasting researches also pay more attention to spatial correlations among the traffic series from different sources (spaces/regions/sensors) except for the temporal correlations. A part of these studies [27, 28, 7, 9] utilize CNN to capture spatial correlations among near regions based on the assumption that traffic series are generated from grid-partitioned cities [28], which does not always hold. To develop more general and widely-used traffic forecasting methods, researchers are shifting to GCN-based models in recent years. These efforts [4, 3, 29, 5, 6, 11, 14, 30, 31] formulate the traffic forecasting problem on graph and utilize the spectral GCN developed in [32, 33] for capturing the prominent spatial interactions among different traffic series. DCRNN [2] re-formulates the spatial dependency of traffic as a diffusion process and extends the previous GCN [32, 33] to a directed graph. Following DCRNN, Graph Wavenet [5] combines GCN with dilated causal convolution networks for saving computation cost in handling long sequence and propose a self-adaptive adaptive adjacency matrix as a complement for the pre-defined adjacent matrix to capture spatial correlations. More recent works such as ASTGCN [6], STSGCN [11] and GMAN [12] further add more complicated spatial and temporal attention mechanisms with GCN to capture the dynamic spatial and temporal correlations. However, these methods can only capture shared patterns among all traffic series and still rely on the pre-defined spatial connection graph.

**Graph Convolutional Networks** GCN [32, 33] is a special kind of CNN generalized for graph-structured data, which is widely used in node classification, link prediction, and graph classification [34]. Most of these works focus on graph representation, which learns node embedding by integrating the features from node's local neighbours based on the given graph structure. To manipulate neighbours' information more accurately, GAT [35] learns to weight the information from different neighbours with attention scores learned by multi-head self-attention mechanism. DIFFPOOL [36] enhances GCN with node clustering to generate hierarchical graph representations. Different from these works dealing with static features, our work deals with dynamically evolving streams and operates on both spatial and temporal dimensions without the given graph structure.

## 3 Methodology

### 3.1 Problem Definition

We target on the multi-step traffic forecasting problem. Consider multitudinous traffic series that contains $N$ correlated univariate time series represented as $\boldsymbol{\mathcal{X}} = \{\boldsymbol{X_{:,0}}, \boldsymbol{X_{:,1}}, ..., \boldsymbol{X_{:,t}}, ...\}$, where $\boldsymbol{X_{:,t}} = \{x_{1,t}, x_{2,t}, ..., x_{i,t}, ...x_{N,t}\}^T \in R^{N \times 1}$ is the recording of $N$ sources at time step $t$, our target is to predict the future values of the correlated traffic series based on the observed historical values. Following in the practice in the time series prediction, we formulate the problem as finding a function $\mathcal{F}$ to forecast the next $\tau$ steps data based on the past $T$ steps historical data:

$$\{\boldsymbol{X_{:,t+1}}, \boldsymbol{X_{:,t+2}}, ..., \boldsymbol{X_{:,t+\tau}}\} = \mathcal{F_\theta}(\boldsymbol{X_{:,t}}, \boldsymbol{X_{:,t-1}}, ..., \boldsymbol{X_{:,t-T+1}}) \tag{1}$$

where $\boldsymbol{\theta}$ denotes all the learnable parameters in the model. In order to accurately manipulate the spatial correlations between different traffic series, the problem is further formulated on graph $\boldsymbol{\mathcal{G}} = (\boldsymbol{\mathcal{V}}, \boldsymbol{\mathcal{E}}, \boldsymbol{A})$, where $\boldsymbol{\mathcal{V}}$ is a set of nodes represent the sources of traffic series and $|\boldsymbol{\mathcal{V}}| = N$, $\boldsymbol{\mathcal{E}}$ is a set of edges, and $\boldsymbol{A} \in R^{N \times N}$ is the adjacent matrix of the graph representing the proximity between nodes or traffic series (e.g., a function of traffic network distance or traffic series similarity). Thus, the problem is modified as:

$$\{\boldsymbol{X_{:,t+1}}, \boldsymbol{X_{:,t+2}}, ..., \boldsymbol{X_{:,t+\tau}}\} = \mathcal{F_\theta}(\boldsymbol{X_{:,t}}, \boldsymbol{X_{:,t-1}}, ..., \boldsymbol{X_{:,t-T+1}}; \boldsymbol{\mathcal{G}}) \tag{2}$$

### 3.2 Node Adaptive Parameter Learning

Most recent work in traffic forecasting deploys GCN to capture the spatial correlations among traffic series and follows the calculations proposed in the spectral domain [32, 33]. According to [33], the graph convolution operation can be well-approximated by $1^{st}$ order Chebyshev polynomial expansion and generalized to high-dimensional GCN as:

$$\boldsymbol{Z} = (\boldsymbol{I_N} + \boldsymbol{D}^{-\frac{1}{2}}\boldsymbol{A}\boldsymbol{D}^{-\frac{1}{2}})\boldsymbol{X}\boldsymbol{\Theta} + \mathbf{b} \tag{3}$$

where $\boldsymbol{A} \in R^{N \times N}$ is the adjacent matrix of the graph, $\boldsymbol{D}$ is the degree matrix, $\boldsymbol{X} \in R^{N \times C}$ and $\boldsymbol{Z} \in R^{N \times F}$ are input and output of the GCN layer, $\boldsymbol{\Theta} \in R^{C \times F}$ and $\mathbf{b} \in R^F$ denote the learnable weights and bias, separately. From the view of one node (e.g., node $i$), the GCN operation can be regarded as transforming the features of node $\boldsymbol{X^i} \in R^{1 \times C}$ to $\boldsymbol{Z^i} \in R^{1 \times F}$ with the shared $\boldsymbol{\Theta}$ and $\mathbf{b}$ among all nodes. While sharing parameters may be useful to learn the most prominent patterns among all nodes in many problems and can significantly reduce the parameter numbers, we find its sub-optimal for traffic forecasting problems. Except for the close spatial correlations between close related traffic series, there also exist diverse patterns among different traffic series due to the dynamic propriety of time series data and various factors of the node that could influence traffic. On the one hand, the traffic streams from two adjacent nodes may also present dissimilar patterns at some particular period because of their specific attributes (e.g., PoI, weather). On the other hand, the traffic series from two disjoint nodes may even show reverse patterns. As a result, only capturing shared patterns among all nodes is not enough for accurate traffic forecasting, and it is essential to maintain a unique parameter space for each node to learn node-specific patterns.

However, assigning parameters for each node will result in $\boldsymbol{\Theta} \in R^{N \times C \times F}$, which is too huge to optimize and would lead to over-fitting problem, especially when $N$ is big. To solve the issue, we propose to enhance traditional GCN with a Node Adaptive Parameter Learning module, which draws insights from the matrix factorization. Instead of directly learning $\boldsymbol{\Theta} \in R^{N \times C \times F}$, NAPL learns two smaller parameter matrix: 1) a node-embedding matrix $\boldsymbol{E_{\mathcal{G}}} \in R^{N \times d}$, where $d$ is the embedding dimension, and $d << N$; 2) a weight pool $\boldsymbol{W_{\mathcal{G}}} \in R^{d \times C \times F}$. Then, $\boldsymbol{\Theta}$ can be generated by $\boldsymbol{\Theta} = \boldsymbol{E_{\mathcal{G}}} \cdot \boldsymbol{W_{\mathcal{G}}}$. From the view of one node (e.g., node $i$), this process extracts parameters $\boldsymbol{\Theta^i}$ for $i$ from a large shared weight pool $\boldsymbol{W_{\mathcal{G}}}$ according to the node embedding $\boldsymbol{E_{\mathcal{G}}^i}$, which can be interpreted as learning node specific patterns from a set of candidate patterns discovered from all traffic series. The same operation can also be used for $\mathbf{b}$. Finally, the NAPL enhanced GCN (i.e., NAPL-GCN) can be formulaed as:

$$\boldsymbol{Z} = (\boldsymbol{I_N} + \boldsymbol{D}^{-\frac{1}{2}} \boldsymbol{A} \boldsymbol{D}^{-\frac{1}{2}}) \boldsymbol{X} \boldsymbol{E_{\mathcal{G}}} \boldsymbol{W_{\mathcal{G}}} + \boldsymbol{E_{\mathcal{G}}} \mathbf{b}_{\mathcal{G}} \tag{4}$$

### 3.3 Data Adaptive Graph Generation

Another problem lies in existing GCN-based traffic forecasting models, which require a pre-defined adjacent matrix $A$ for the graph convolution operation. Existing work mainly utilizes distance function or similarity metrics to calculate the graph in advance. There are mainly two approaches for defining $A$: 1) distance function, which defines the graph according to the geographic distance among different nodes[2, 4]; 2) similarity function, which defines the node proximity by measuring the similarity of the node attributes (e.g., PoI information) [7, 14] or traffic series itself [3]. However, these approaches are quite intuitive. The pre-defined graph cannot contain complete information about spatial dependency and is not directly related to prediction tasks, which may result in considerable biases. Besides, these approaches cannot be adapted to other domains without appropriate knowledge, making existing GCN-based models ineffective.

To solve the issue, we propose a Data Adaptive Graph Generation (DAGG) module to infer the hidden inter-dependencies from data automatically. The DAGG module first randomly initialize a learnable node embedding dictionaries $\boldsymbol{E_A} \in R^{N \times d_e}$ for all nodes, where each row of $\boldsymbol{E_A}$ represents the embedding of a node and $d_e$ denotes the dimension of node embedding. Then, similar as defining the graph by nodes similarity, we can infer the spatial dependencies between each pair of nodes by multiplying $\boldsymbol{E_A}$ and $\boldsymbol{E_A^T}$:

$$\boldsymbol{D}^{-\frac{1}{2}} \boldsymbol{A} \boldsymbol{D}^{-\frac{1}{2}} = softmax(ReLU(\boldsymbol{E_A} \cdot \boldsymbol{E_A^T})) \tag{5}$$

where $softmax$ function is used to normalize the adaptive matrix. Here, instead of generating $\boldsymbol{A}$ and calculating a Laplacian matrix, we directly generate $\boldsymbol{D}^{-\frac{1}{2}} \boldsymbol{A} \boldsymbol{D}^{-\frac{1}{2}}$ to avoid unnecessary and repeated calculations in the iterative training process. During training, $\boldsymbol{E_A}$ will be updated automatically to learn the hidden dependencies among different traffic series and get the adaptive matrix for graph convolutions. Comparing with the self-adaptive adjacent matrix in [5], DAGG module is simpler and the learned $\boldsymbol{E_A}$ has better interpret-ability. Finally, the DAGG enhanced GCN can be formulated as:

$$\boldsymbol{Z} = (\boldsymbol{I_N} + softmax(ReLU(\boldsymbol{E_A} \cdot \boldsymbol{E_A^T}))) \boldsymbol{X} \boldsymbol{\Theta} \tag{6}$$

When dealing with extremely large graphs (i.e., $N$ is huge), DAGG may require heavy computation cost. Graph partition and sub-graph training methods [12, 37] could be applied to address the problem.

### 3.4 Adaptive Graph Convolutional Recurrent Network

Except for the spatial correlations, traffic forecasting also involves complex temporal correlations. In this part, we introduce an Adaptive Graph Convolutional Recurrent Network (AGCRN), which integrates NAPL-GCN, DAGG, and Gated Recurrent Units (GRU) to capture both node-specific spatial and temporal correlations in traffic series. AGCRN replaces the MLP layers in GRU with our NAPL-GCN to learn node-specific patterns. Besides, it discoveries spatial dependencies automatically with the DAGG module. Formally:

$$\begin{aligned}
\widetilde{\boldsymbol{A}} &= softmax(ReLU(\boldsymbol{E}\boldsymbol{E^T})) \\
\boldsymbol{z_t} &= \sigma(\widetilde{\boldsymbol{A}}[\boldsymbol{X_{:,t}}, \boldsymbol{h_{t-1}}]\boldsymbol{E}\boldsymbol{W_z} + \boldsymbol{E}\boldsymbol{b_z}) \\
\boldsymbol{r_t} &= \sigma(\widetilde{\boldsymbol{A}}[\boldsymbol{X_{:,t}}, \boldsymbol{h_{t-1}}]\boldsymbol{E}\boldsymbol{W_r} + \boldsymbol{E}\boldsymbol{b_r}) \\
\hat{\boldsymbol{h}}_t &= tanh(\widetilde{\boldsymbol{A}}[\boldsymbol{X_{:,t}}, \boldsymbol{r} \odot \boldsymbol{h_{t-1}}]\boldsymbol{E}\boldsymbol{W_{\hat{h}}} + \boldsymbol{E}\boldsymbol{b_{\hat{h}}}) \\
\boldsymbol{h_t} &= \boldsymbol{z} \odot \boldsymbol{h_{t-1}} + (1 - \boldsymbol{z}) \odot \hat{\boldsymbol{h}}_t
\end{aligned} \tag{7}$$

where $\boldsymbol{X_{:,t}}$ and $\boldsymbol{h_t}$ are input and output at time step $t$, $[\cdot]$ denotes the concate operation, $\boldsymbol{z}$ and $\boldsymbol{r}$ are reset gate and update gate, respectively. $\boldsymbol{E}$, $\boldsymbol{W_z}$, $\boldsymbol{W_r}$, $\boldsymbol{W_{\hat{h}}}$, $\boldsymbol{b_z}$, $\boldsymbol{b_r}$, and $\boldsymbol{b_{\hat{h}}}$ are learnable parameters in AGCRN. Similar to GRU, all the parameters in AGCRN can be trained end-to-end with back-propagation through time. As can be observed from the equation, AGCRN unifies all the embedding matrix to be $\boldsymbol{E}$ instead of learning separate node embedding matrix in different NAPL-GCN layers and DAGG. This gives a strong regularizer to ensure the nodes embedding consistent among all GCN blocks and gives our model better interpretability.

### 3.5 Multi-step traffic prediction

To achieve multi-step traffic prediction, we stack several AGCRN layers as an encoder to capture the node-specific spatial-temporal patterns and represents the input (i.e., historical data) as $H \in R^{N \times d_o}$. Then, we can directly obtain the traffic prediction for the next $\tau$ steps of all nodes by applying a linear transformation to project the representation from $R^{N \times d_o}$ to $R^{N \times \tau}$. Here, we do not generate the output in the sequential manner as it would increase the time consumption significantly.

We choose L1 loss as our training objective and optimize the loss for multi-step prediction together. Thus, the loss function of AGCRN for multi-step traffic prediction can be formulated as:

$$\mathcal{L}(\boldsymbol{W_\theta}) = \sum_{i=t+1}^{i=t+\tau} |\boldsymbol{X_{:,i}} - \boldsymbol{X'_{:,i}}| \tag{8}$$

where $\boldsymbol{W_\theta}$ represents all the learnable parameters in the network, $\boldsymbol{X_{:,i}}$ is the ground truth, and $\boldsymbol{X'_{:,i}}$ is the prediction of all nodes at time step $i$. The problem can be solved via back-propagation and Adam optimizer.

## 4 Experiments

### 4.1 Datasets

To evaluate the performance of our work, we conduct experiments on two public real-world traffic datasets: PeMSD4 and PeMSD8 [6, 11]. PeMS means Caltrans Performance Measure System (PeMS) [38], which measures the highway traffic of California in real-time every 30 seconds.

**PeMSD4**: The PeMSD4 dataset refers to the traffic flow data in the San Francisco Bay Area. There are 307 loop detectors selected within the period from 1/Jan/2018 to 28/Feb/2018.

**PeMSD8**: The PeMSD8 dataset contains traffic flow information collected from 170 loop detectors on the San Bernardino area from 1/Jul/2016 - 31/Aug/2016.

**Data Preprocess:** The missing values in the datasets are filled by linear interpolation. Then, both datasets are aggregated into 5-minute windows, resulting in 288 data points per day. Besides, we normalize the dataset by standard normalization method to make the training process more stable. For multi-step traffic forecasting, we use one-hour historical data to predict the next hour's data, i.e.,

we organize 12 steps' historical data as input and the following 12 steps data as output. We split the datasets into training sets, validation sets, and test sets according to the chronological order. The split ratio is 6:2:2 for both datasets. Although our method does not need a pre-defined graph, we use the pre-defined graph for our baselines. Detailed dataset statistics are provided in the appendix.

## 4.2 Experimental Settings

To evaluate the overall performance of our work, we compare AGCRN with widely used baselines and state-of-the-art models, including 1) Historical Average (HA): which models the traffic as a seasonal process and uses the average of previous seasons (e.g., the same time slot of previous days) as the prediction; 2) Vector Auto-Regression (VAR) [39]: a time series model that captures spatial correlations among all traffic series; 3) GRU-ED: an GRU-based baseline and utilize the encoder-decoder framework [40] for multi-step time series prediction; 4) DSANet [41]: a correlated time series prediction model using CNN networks for capturing temporal correlations with one time-series and self-attention mechanism for spatial correlations; 5) DCRNN [2]: diffusion convolution recurrent neural network, which formulates the graph convolution with the diffusion process and combines GCN with recurrent models in an encoder-decoder manner for multi-step prediction; 6) STGCN [4]: a spatio-temporal graph convolutional network that deploys GCN and temporal convolution to capture spatial and temporal correlations, respectively; 7) ASTGCN [6]: attention-based spatio-temporal graph convolutional network, which further integrates spatial and temporal attention mechanisms to STGCN for capturing dynamic spatial and temporal patterns. We take its recent components to ensure the fairness of comparison; 8) STSGCN [11]: Spatial-Temporal Synchronous Graph Convolutional Network that captures spatial-temporal correlations by stacking multiple localized GCN layers with adjacent matrix over the time axis.

All the deep-learning-based models, including our AGCRN, are implemented in Python with Pytorch 1.3.1 and executed on a server with one NVIDIA Titan X GPU card. We optimize all the models by Adam optimizer for a maximum of 100 epochs and use an early stop strategy with the patience of 15. The best parameters for all deep learning models are chosen through a carefully parameter-tuning process on the validation set.

## 4.3 Overall Comparison

We deploy three widely used metrics - Mean Absolute Error (MAE), Root Mean Square Error (RMSE), and Mean Absolute Percentage Error (MAPE) to measure the performance of predictive models. Table 1 presents the overall prediction performances, which are the averaged MAE, RMSE and MAPE over 12 prediction horizons, of our AGCRN and eight representative comparison methods. We can observe that: 1) GCN-based methods outperform baselines and self-attention-based DSANet, demonstrating the importance of modeling spatial correlations explicitly and the effectiveness of GCN in traffic forecasting; 2) our method further improves GCN-based methods with a significant margin. AGCRN brings more than 5% relative improvements to the existing best results in MAE and MAPE for both PeMSD4 and PeMSD8 dataset. Fig. 2 further shows the prediction performance at each horizon in the PeMSD4 dataset. AGCRN balances short-term and long-term prediction well and achieves the best performance for almost all horizons (except for the first step). Besides, the performance of AGCRN deteriorate much slower than other GCN-based models (see appendix for similar results in the PeMSD8 dataset).

Overall, the results demonstrate that AGCRN can accurately capture the spatial and temporal correlations in the correlated traffic series and achieve promising predictions.

## 4.4 Ablation Study

To better evaluate the performance of NAPL and DAGG, we conduct a comprehensive ablation study. The baseline for our ablation study is GCGRU, which integrates traditional GCN with GRU to capture spatial and temporal correlations. We construct NAPL-GCGRU by replacing traditional GCN with our NAPL-GCN and DAGG-GCGRU by replacing the pre-defined graph with the DAGG module. AGCCRN-I is the variant of our AGCRN, which does not unify the node embeddings but employs an independent node embedding matrix among different NAPL-GCN layers and DAGG. The experiments on the PeMSD4 dataset are illustrated in Fig. 3. We can observe that: 1) NAPL-GCGRU generally outperforms GCGRU and AGCRN-I outperforms DAGG-GCGRU, demonstrating the

Table 1: Overall prediction performance of different methods on the PeMSD4 dataset and PeMSD8 dataset, results with * are reported performance in the paper used the same datasets and results with __ are the best performance achieved by baselines. (smaller value means better performance)

| Model | Dataset | PeMSD4 | | | PeMSD8 | | |
|---|---|---|---|---|---|---|---|
| | Metrics | MAE | RMSE | MAPE | MAE | RMSE | MAPE |
| HA | | 38.03 | 59.24 | 27.88% | 34.86 | 52.04 | 24.07% |
| VAR | | 24.54 | 38.61 | 17.24% | 19.19 | 29.81 | 13.10% |
| GRU-ED | | 23.68 | 39.27 | 16.44% | 22.00 | 36.23 | 13.33% |
| DSANet [41] | | 22.79 | 35.77 | 16.03% | 17.14 | 26.96 | 11.32% |
| DCRNN [2] | | 21.22 | 33.44 | 14.17% | 16.82 | 26.36 | 10.92% |
| STGCN [4] | | 21.16 | 34.89 | 13.83% | 17.50 | 27.09 | 11.29% |
| ASTGCN [6] | | 22.93 | 35.22 | 16.56% | 18.25 | 28.06 | 11.64% |
| STSGCN [11] | | 21.19* | 33.65* | 13.90%* | 17.13* | 26.86* | 10.96%* |
| AGCRN (ours) | | 19.83 | 32.26 | 12.97% | 15.95 | 25.22 | 10.09% |
| Improvements | | +6.29% | +3.52% | +6.22% | +5.17% | +4.32% | +7.60% |

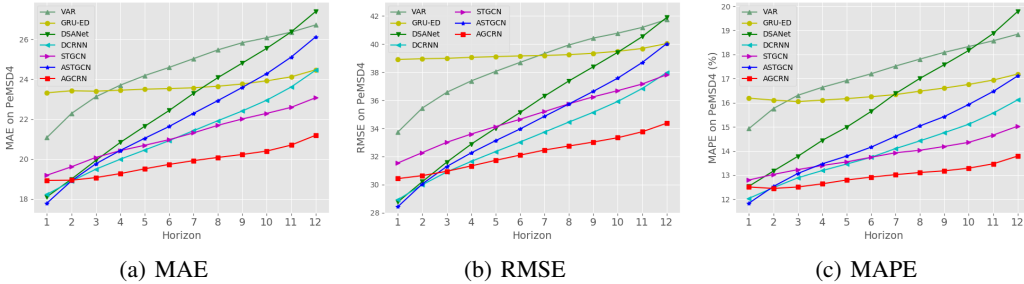

(a) MAE          (b) RMSE          (c) MAPE

Figure 2: Prediction performance comparison at each horizon on the PeMSD4 dataset.

necessity of capturing node-specific patterns. Moreover, NAPL mainly enhances the long-term (e.g., 30Min and 60 Min) prediction but slightly harms the short-term (e.g., 5Min and 15 Min) prediction. We conjecture the reason is that long-term prediction lacks enough useful information from historical observations and thus benefits from the specific node embedding learned by the NAPL module to deduce future patters. At the same time, short-term prediction can obtain enough information from historical observations. 2) DAGG-GCGRU improves GCGRU, and AGCRN-I beats NAPL-GCGRU. Both demonstrate the superiority of DAGG in inferring spatial correlations. The results also indicate that GCN-based methods can potentially be applied to more general correlated time series forecasting tasks with the help of our DAGG module, and pre-defining an adjacent matrix is not necessary; 3) AGCRN achieves the best performance, demonstrating that we can share the node embedding among all the modules and learn a unified node embedding for each node from the data.

Overall, our NAPL and DAGG modules can be deployed either separately and jointly, and they consistently boost the prediction performance.

### 4.5 Model Analysis

**Graph Generation** To further investigate DAGG, we compare it with two variants: 1) DAGG-r, which removes the identity matrix in Eq. 6; 2) DAGG-2 which mimics the second-order Chebyshev polynomial expansion in GCN [4, 33] with our learned $D^{-\frac{1}{2}}AD^{-\frac{1}{2}}$. The backbone network is AGCRN-I, which does not share the embedding matrix among NAPL-GCN and DAGG to avoid the constraints from the NAPL module. As shown in Table 2 (where DAGG-1 follows Eq. 6), removing the identity matrix from DAGG significantly harms the prediction performance, which presents the

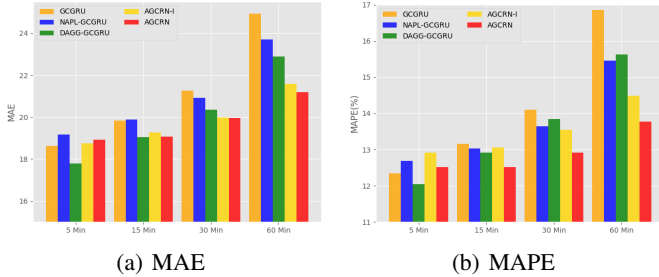

(a) MAE           (b) MAPE

Figure 3: Ablation study on the PeMSD4 dataset.

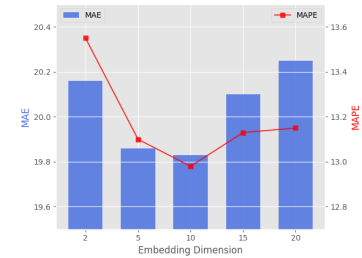

Figure 4: Influence of the embedding dimension.

importance of highlighting the self-information manually in prediction. Besides, DAGG-2 achieves similar performance with DAGG-1, which is consistent with the existing works [33, 4, 2] using pre-defined graphs. The results reveal that the generated graph Laplacian matrix $D^{-\frac{1}{2}}AD^{-\frac{1}{2}}$ shares similar property as the pre-defined graph in Chebyshev polynomial expansion.

Table 2: Analysis of graph generation process on the PeMSD4 dataset.

| Model | 15 Min | | | 60 Min | | | Average | | |
|---|---|---|---|---|---|---|---|---|---|
| | MAE | RMSE | MAPE | MAE | RMSE | MAPE | MAE | RMSE | MAPE |
| DAGG-r | 21.85 | 35.03 | 14.96% | 26.54 | 41.07 | 17.91% | 23.35 | 37.07 | 15.82% |
| DAGG-1 | 19.15 | 30.65 | 13.15% | 21.98 | 34.91 | 14.82% | 20.18 | 32.30 | 13.70% |
| DAGG-2 | 19.26 | 31.20 | 13.06% | 21.58 | 34.73 | 14.49% | 20.11 | 32.56 | 13.58% |

**Embedding Dimension** One key parameter in AGCRN is the dimensions of the node embedding, which not only influences the quality of the learned graph but also decides the parameter diversity in NAPL-GCN layers. Fig. 4 shows the effects of different embedding dimensions to AGCRN on the PeMSD4 dataset. AGCRN obtains relatively good performance for all the tested embedding dimensions, which shows the robustness of our methods. Besides, AGCRN achieves the best performance when the embedding dimension is set to 10. Both an excessively small and large node embedding dimension will lead to weaker performance. On the one hand, node embedding with a larger dimension can contain more information and thus help our DAGG module to deduce more accurate spatial correlations. On the other hand, a larger node embedding dimension will significantly increase the parameter numbers in the NAPL module, making the model harder to optimize and causing over-fitting. Overall, it would be a good practice for AGCRN to find a suitable node embedding dimension and balance the model's performance and complexity.

**Computation Cost** To evaluate the computation cost, we compare the parameter numbers and training time of AGCRN with DCRNN, STGCN, and ASTGCN on the PeMSD4 dataset in Table 3. When he node embedding dimension is set to 10, AGCRN has five times more parameters than the DCRNN model as a sacrifice for learning node-specific patterns. In terms of the training time, AGCRN runs slightly faster than DCRNN as we generate all predictions directly instead of the iterative manner in DCRNN. STGCN is the fastest thanks to the temporal convolution structure. However, it will require more parameters and training time to add spatial and temporal attention mechanisms to STGCN for learning more accurate spatial-temporal patterns (e.g., ASTGCN). Considering the significant performance improvement (as shown in Table 1), the computation cost of AGCRN is moderate.

## 5 Discussion

Multivariate/correlated time series prediction is a fundamental task for many applications, such as epidemic transmission forecasting [42], meteorology (e.g., air quality, rainfall) prediction [43], stock forecasting [44], and sale prediction [45]. While our work is motivated by the traffic forecasting task, the proposed two adaptive modules and our AGCRN model may also be adapted to a wide variety of multivariate/correlated time series predictive tasks separately or jointly. It is possible to automatically

Table 3: The computation cost on the PeMSD4 dataset, "dim" means the dimension of $E$.

| Model | # Parameters | Training Time (epoch) |
|---|---|---|
| DCRNN | 149057 | 36.39 s |
| STGCN | 211596 | 16.36 s |
| ASTGCN | 450031 | 49.47 s |
| AGCRN (dim=2) | 150386 | 33.88 s |
| AGCRN (dim=10) | 748810 | 35.56 s |

discover the inter-dependency among different correlated series from data, which bridges the gap between graph-based prediction models and general correlated time series forecasting problems that cannot pre-define the graph easily. Our future work will focus on examining the scale-ability of our work from two perspectives: 1) data perspective - validating the performance of AGCRN on more time series prediction tasks; 2) model perspective - adapting NAPL and DAGG to more GCN-based traffic forecasting models.

## 6   Conclusion

In this paper, we propose to enhance the traditional graph convolutional network with node adaptive parameter learning and data-adaptive graph generation modules for learning node-specific patterns and discovering spatial correlations from data, separately. Based on the two modules, we further propose the Adaptive Graph Convolutional Recurrent Network, which can capture node-specific spatial and temporal correlations in time-series data automatically without a pre-defined graph. Extensive experiments on multi-step traffic forecasting tasks demonstrate the effectiveness of both AGCRN and the proposed adaptive modules. This work sheds light on applying GCN-based models in correlated time series forecasting by inferring the inter-dependency from data and reveals that learning node-specific patterns is essential for understanding correlated time series data.

## Broader Impact

In general, this work enables more accurate traffic forecasting, which facilities the higher-lever traffic scheduling such as taxi dispatch and route planing. In this way, our work can help save time for travelers, improve efficiency and income for transport operators, and save energy consumption. In a broad sense, adaptability is desirable in correlated time series analysis for broad social and business applications in the era of big data. The proposed adaptive modules enable elevated robustness of data analysis and relevant applications based on dynamic, interdependent, time-series data. This research generally supports better modeling and analysis of multiple channels of data based on graph structures with complex explicit and implicit correlations. It has implications and potentially accelerates the research progress in address many world-scale economic and societal issues that rely on complex times series data, such as predictions of influenza outbreak, economic growth, and climate change. A potential negative impact of this work is the fairness problem in the ride-sharing platforms. In the case that cabs supply cannot guarantee demand, platforms may emphasize the predicted high-demand areas too much, which would increase the waiting time of travelers in the low-demand areas.

## Footnotes

[1]Code available at: https://github.com/LeiBAI/AGCRN

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
