[Supplementary Material · Appendix.pdf]

# A Appendix

To support reproducibility of the results in this paper, we have submitted our code and datasets as the supplementary information. Here, we will present the datasets statistics, evaluation metrics, implementation details, and more results.

## A.1 Datasets Statistics

The dataset used in our experiments (namely PeMSD4 dataset and PeMSD8 dataset) contain the traffic flow data measured by road traffic sensors. As introduced in Section 3.1, we formulate the traffic forecasting problem on a graph where each node corresponds to a traffic sensor. Our ASTGCN can infer spatial proximity from data by DAGG module automatically. Thus is does not require pre-defining the adjacent matrix. For graph-based baselines, we reuse the pre-defined graph given in [11] to capture spatial correlations. The connectivity between different nodes is determined by the actual road network. If two monitors are on the same road, then they are considered connected. The statistics about the two datasets are shown in Table 4.

Table 4: Summary statistics of the PeMSD4 and PeMSD8 dataset

| Dataset | Time Span | #Nodes | #Edges | #Samples | Data Range | Median |
|---------|-----------|--------|--------|----------|------------|--------|
| PeMSD4 | 1/Jan/2018 - 28/Feb/2018 | 307 | 340 | 16992 | $0 \sim 919$ | 180 |
| PeMSD8 | 1/Jul/2016 - 31/Aug/2016 | 170 | 277 | 17856 | $0 \sim 1147$ | 215 |

## A.2 Evaluation Metrics

We use three evaluation metrics to measure the performance of predictive models. Let $\boldsymbol{X}_{:,i} \in R^{N \times 1}$ be the ground truth traffic of all nodes at time step $i$, $\boldsymbol{X}'_{:,i} \in R^{N \times 1}$ be the predicted values, and $\Omega$ be indices of observed samples. The metrics are defined as follows.

Mean Absolute Error (MAE)

$$MAE = \frac{1}{|\Omega|} \sum_{i \in \Omega} |\boldsymbol{X}_{:,i} - \boldsymbol{X}'_{:,i}|$$

Root Mean Square Error (RMSE)

$$RMSE = \sqrt{\frac{1}{|\Omega|} \sum_{i \in \Omega} (\boldsymbol{X}_{:,i} - \boldsymbol{X}'_{:,i})^2}$$

Mean Absolute Percentage Error (MAPE)

$$MAPE = \frac{1}{|\Omega|} \sum_{i \in \Omega} \left| \frac{\boldsymbol{X}_{:,i} - \boldsymbol{X}'_{:,i}}{\boldsymbol{X}_{:,i}} \right|$$

## A.3 Implementation Details

The details of the baselines are as follows:

- HA: the historical average model operates on each traffic series separately, and it averages all the historical traffic at the same time slot to predict current traffic. Historical Average does not depend on recent data and thus the performance is invariant for 12 forecasting horizons.

- VAR: we implement the VAR model based on *statsmodel* python package and search the number of lags among {1, 3, 6, 9, 12}. The number of lags is set to 12 for both PeMSD4 and PeMSD8 datasets.

- GRU-ED: we implement an encoder-decoder model based on GRU with Pytorch. GRU-ED contains two layers of GUR for both encoder and decoder; each layer has 128 hidden units. A fully-connected layer projects the output of the decoder at each time step to a prediction. We set the batch size to 64, learning rate to 0.001, and the loss function to L1 when training the model.

- DSANet: we reuse the code released in the original paper and tune the parameters carefully for our dataset according to the validation error. We set the CNN filter size to 3, number of CNN kernels to 64, number of attention blocks to 3, dropout probability to 0.1, and the learning rate to 0.001.

- DCRNN: similar to GRU-ED, the DCRNN model also deploys the ecoder-decoder framework for multi-step traffic forecasting. It contains two-layers DCGRU for both encoder and decoder. We set the number of GRU hidden units to 64, the maximum step of randoms walks to 3, the initial learning rate to 0.01. We decrease the learning rate tby $\frac{1}{10}$ every 20 epochs starting from $10_{th}$ epochs.

- STGCN: STGCN contains two spatial-temporal convlutional blocks, one temporal convolutional layer and one output layer. Different from the original STGCN, we implement the output layer to generate prediction for all horizons at one time (instead of one step per time). Following the practice of STGCN, we set the size of temporal kernel to 2, the order of Chebyshev polynomials to 1, and the filter number to 64 for both CNN and GCN. Besides, We set the learning rate to 0.003 for the PeMSD4 dataset and 0.001 for the PeMDS8 dataset.

- ASTGCN: The orginal ASTGCN model ensembles three bolocks to process the recent, daily-periodic, and weekly-periodic segments for capturing multi-scale temporal correlations. We take its recent component that only uses recent input segments for a fair comparison. For implementation, we reuse the code and parameters released in the original paper and train the model with a L1 loss function.

- STSGCN: We reuse the results reported in the original paper directly for our overall comparison as it conducts experiments on the PeMSD4 and PeMSD8 datasets with the same evaluation metrics.

**AGCRN**: Our model stacks two layers AGCRN to capture the node-specific spatial and temporal dynamics. The output at the last step is used as the representation of the historical traffic series, which is directly mapped to the predictions for all horizons by linear transformation . For the hype-parameters, we set the hidden unit to 64 for all the AGCRN cells and the batch size also to 64. We search the learning rate among {0.0007, 0.001, 0.003, 0.005, 0.009}, the embedding dimension among {1, 3, 5, 10, 15, 20, 30} for the PeMSD4 dataset and among {1, 2, 3, 5, 8, 10, 15} for the PeMSD8 dataset. Finally, the learning rate is set to 0.003 for both datasets, and the embedding dimension is to 10 for the PeMSD4 dataset and 2 for the PeMSD8 dataset. Besides, we choose L1 Loss as the loss function and do not use any non-mentioned optimization tricks such as learning rate decay, weights decay, or gradient normalization when training our model.

For all the deep learning models, we optimize them with the Adam optimizer for 100 epochs and use an early stop strategy with the patience of 15 by monitoring the loss in the validation set.

### A.4 Multi-step Prediction on PeMSD8

|(a) MAE|(b) RMSE|(c) MAPE|

Figure 5: Prediction performance comparison at each horizon on the PeMSD8 dataset.

Fig. 5 presents the prediction performance of our AGCRN and baselines at each horizon on the PeMSD8 dataset. STSGCN is not included because the step-wise results of it are not reported in [11]. Besides, we omit HA as it's performance is consistent for all 12 horizons. Our AGCRN model outperforms existing baselines with a significant margin, especially for long-term predictions. Besides, the performance of AGCRN deteriorates much slower than the other GCN-based models. The observations are similar on the PeMSD4 dataset.

## A.5 Prediction Visualization

(a)

(b)

(c)

Figure 6: Traffic forecasting visualization.

(a)

(b)

(c)

Figure 7: Traffic forecasting visualization.