[Reviews · NeurIPS 2020]

Review 1

Summary and Contributions: This paper proposes a GNN based model for traffic forecasting. Its backend model is a recurrent GNN model for sequential data. GNN is used to capture structure information between multiple time series and recurrent solution is used to adopt temporal modeling. The key contribution for this work lies in the first part, which can be summarized as follows 1. Different from general GNN that uses a shared model to learn embedding for all nodes, this model hints using unique parameters for each node. 2. Once the adjacency matrix is not provided, it introduces learnable parameters for each node, and its pairwise similarly are used to formulize the adjacency matrix. Such approach can be trained end-to-end.

Strengths: 1. This paper is well motivated. Indeed for time series data, especially those from different resources, doing unique fitting for each data is reasonable. This paper further has a low rank assumption for time series data, which is a reasonable set up. 2. This papers provides a straight forward and easy to implement solution for the adjacency matrix when the graph structure is unknown.

Weaknesses: It is not totally clearly where the superiority is coming from and it is not very convincing whether the experiments are fair comparisons. a. There is no regularization for the E_g to be orthogonal, where is not very clear what is the underlying structure of E_g. Intuitively, a in-detail analysis should be provided, e.g, What is the rank of E_g, especially when sparse regularization is introduced. b. Since E_a has parameter in size of Nxd_c, I am in doubt whether this model is over-parameterized for the embedding. It is important to compare the model size of the proposed one and those baselines; further, the analysis for the learned embeddings are missing. It would be interesting to understand what types of time series are picked to be connected. What is the relation between such time series, are they only highly correlated or do they have strong lead-lag relation or whatever? How could be model achieve sparsity in the graph? It is common some other time series are not informative to forecast an target time series, if so, how to avoid such link in the graph? Some insightful and essential analysis are missing.

Correctness: The claims and method are correct

Clarity: This paper is clearly writtten and easy to follow

Relation to Prior Work: This paper has properly refer previous relatede work.

Reproducibility: Yes

Additional Feedback:


Review 2

Summary and Contributions: This paper investigates the problem of traffic flow forecasting, and proposes methods to capture node specific patterns as well as learn the inter-dependencies among nodes without a predefined graph. The investigated problem is important, and the empirical results on two real-world datasets show very promising results. -- Update after rebuttal: Thanks for the feedback. It will be helpful to add discussion in the paper about (1) Integrates existing graph structure information. (2) Multiple step prediction. (3) and scalability.

Strengths: S1: The investigated problem is important, and empirical results on real-world datasets are promising. S2: The idea of using learning node-specific parameters with shared basis (or lower rank decomposition) is very interesting. S3: The idea of learning the normalized adjacency matrix from scratch with random embedding is very interesting. Here E (E_G or E_A) can be seen the representation of the node, W (W_z, W_r or W_h) can be seen as transformations from the node embedding space to target spaces S4: The proposed approaches do not seem to limit to traffic flow prediction, and can be potentially applied to a much broader domain wherever pair-wise correlation is important.

Weaknesses: W1: The presentation can be improved. It seems that this paper is finished in a rush (or has last minute change). - For example, in Table 1, Metr-LA is shown, while it is actually means PeMSD8. - In Table 1, "*" seems to be missing from? It seems that the paper originally compares with baselines on METR-LA (citing performance from other paper), but decide to switch to PeMSD8 for some reason. - In Table 2, the model names are NAGG-x while the ones in the corresponding paragraphs are DAGG-x. W2: The experimental are not quite convincing. - In Table 1, the "*"sign is not present in any numbers, does it mean all the baseline methods in the Table implemented by the author? - Most baselines, e.g., DCRNN, STGCN, ASTGCN, have their performance evaluated on PeMSD4, PeMSD7, PeMSD8, METR-LA, PEMS-BAY (w.r.t. traffic speed rather than traffic flow). Thus, it will be more convincing to compare with baselines with numbers achieved in the corresponding papers rather than regenerating the results on a different setting. - It will be very helpful to see the comparison on the Metr-LA dataset (it seems the lib/dataloadr.py also has METR-LA as an option) It will be helpful to show whether the proposed approach still able to achieve superior performance. W3: More justification of the proposed approach. - Learning graph structure. It is very surprising to see that the proposed approach achieves better results than baselines without using graph information. Does it mean the pair-wise distance or similarity cannot provide any extra information? It is also interesting to see if it is possible to achieve improved result if these information is utilized in AGCRN (e.g., as regularization of the embedding). - Multiple step prediction. AGCRN generates the output in one-shot, instead of sequential manner (without positional embedding). It will be interesting to see if sequentially generates the output or adding positional embedding will further improve the result. - It seems that the proposed method need O(N^2) computation, comparing to some baseline whose time complexity is O(|E|) where |E| is the number of edges. It will be helpful to discuss how to scale the AGCRN to a large scale graphs. Will graph partition applicable? Will mini-batch training applicable?

Correctness: Both the idea of using learning node-specific parameters with shared basis (or lower rank decomposition) and the idea of learning the normalized adjacency matrix from scratch with random embedding is intuitive. Besides, the experimental results in Table 1, 2 also support the claims in the paper.

Clarity: The paper is generally easy to follow, and the proposed approach is described in a relatively clear way. However, it seems that this paper is finished in a rush (or has last minute changes). - For example, in Table 1, Metr-LA is shown, while it is actually means PeMSD8. - In Table 1, "*" seems to be missing from? It seems that the paper originally compares with baselines on METR-LA (citing performance from other paper), but decide to switch to PeMSD8 for some reason. - In Table 2, the model names are NAGG-x while the ones in the corresponding paragraphs are DAGG-x.

Relation to Prior Work: Yes, important related works are discussed and differentiated in Section 2.

Reproducibility: Yes

Additional Feedback:


Review 3

Summary and Contributions: This paper proposes two adaptive modules: 1) Node Adaptive Parameter Learning (NAPL) 2) Data Adaptive Graph Generation (DAGG) to improve the capability of traditional Graph Neural Networks. Further, they combine both modules to propose a new variant of Graph Neural Networks. The experiments show some promising results for proposed methods on two traffic datasets.

Strengths: 1. Two proposed modules can work separately or jointly to improve the traditional graph neural network. 2. The evaluation results are solid and the improvements are significant. 3. The proposed framework can be extended to other time-series applications.

Weaknesses: 1. In section 3.5, I wonder why the authors particularly choose L1 loss to optimize and why not using other loss functions, e.g., L2 loss (square loss). Can the authors elaborate more on this? 2. In section 4.1, the authors formulate the problem as a 12-step prediction. Does it suggest that 0-12 hrs are inputs and 13-24 hrs are targets? 3. In section 4.4, I am a little bit confused about the setup for short-term (e.g., 5mins) and long-term predictions (e.g., 60 mins). Are they also multi-step predictions? I would like the authors to clarify this part.

Correctness: The proposed method looks correct and the evaluation results seem reasonable to me.

Clarity: This paper is well-written and easy to follow.

Relation to Prior Work: The authors have demonstrated the differences between the proposed work and previous works.

Reproducibility: Yes

Additional Feedback: Please try to address my concerns in "Weaknesses". ==============post-rebuttal============== Thanks for the response. I will keep my previous rating after reading the rebuttal.


Review 4

Summary and Contributions: This paper proposed a deep learning model Adaptive Graph Convolutional Recurrent Network (AGCRN) for traffic forecasting while the pre-defined graph is avoidable. The proposed model consisted of three main parts: 1) a Node Adaptive Parameter Learning (NAPL) module to capture node-specific patterns; 2) a Data Adaptive Graph Generation (DAGG) module to infer the inter-dependencies among different traffic series automatically. 3) recurrent network. AGCRN captured fine-grained spatial and temporal correlations in traffic series automatically and showed good performance on two real-world traffic datasets.

Strengths: Existing GCN-based methods require pre-defining an inter-connection graph by similarity or distance measures to capture the spatial correlations. That further requires substantial domain knowledge and is sensitive to the graph quality. This paper proposed a GCN-based traffic forecasting model while the pre-defined graph is avoidable. Experiments on two real-world traffic datasets show the proposed model outperformed state-of-the-art models by a significant margin without pre-defined graphs about spatial connections.

Weaknesses: 1. Relationship to previous work. First and foremost, the paper lacks a discussion on previous related work on graph convolution network (GCN). As per my knowledge, the technologies used in 3.2 NAPL and 3.3 DAGG have already been proposed/discussed in [1, 2, 3]. These papers are very famous for improving GCN, but not mentioned in this paper. As a reader, I am not sure what contributions exactly this paper has made. I encourage the authors to present which papers the technologies that they use to improve GCN are from, and what this paper is adding to the literature or how this paper modifying these technologies to adapt traffic forecasting task. [1] Graph Attention Network, ICLR 2018. [2] Self-Attention Graph Pooling, ICML 2019. [3] Hierarchical Graph Representation Learning with Differentiable Pooling, NeurIPS 2018. 2. Technical Novelty: as explained in first item, due to lacks of a discussion on previous related work on graph convolution network, the proposed model in this paper seems replace the GCN part of previous traffic forecasting models, with a more advanced existing one. The technical novelty of the paper is limited. But I understand that technical novelty is not the aim of the paper, so I list this item in the bottom.

Correctness: The claims and method in this paper seem correct. The experimental settings seem reasonable. Code for the model was submitted.

Clarity: This paper is well written.

Relation to Prior Work: This paper clearly discussed how this work differs from previous traffic forecasting work. But it lacks a discussion on previous related work on graph convolution network (GCN).

Reproducibility: Yes

Additional Feedback: 1. What's Metr-LA in table 1? Metr-LA has not been introduced in the paper. 2. STSGCN is shown from reference [4], which should be from refernce [11].

[Author Response · NeurIPS 2020]

**To Reviewer 1:** Thanks for your review. (I) Fair comparisons: the datasets in our evaluation come from ASTGCN and STSGCN. Our results can be cross-checked to examine the performance (e.g., compare ref [11]). The code and datasets are also provided for reproducing our results. For other baselines (e.g., DCRNN, STGCN), we fine-tuned them to choose the best settings and hype-parameters to ensure our comparison is fair. We believe our results are solid and the superiority can be explained from two views: first, the predefined graph, which is provided by ASTGCN and STSGCN, may be sub-optimal compared to our adaptively learned graph; second, the NAPL module can do unique fitting for each node effectively. (II) Parameter size: we compared the number of parameters in Table 3 (page 8), which shows that AGCRN has moderate parameters when compared with baselines (e.g., ASTGCN). Besides, if setting the embedding dimension to a smaller value, AGCRN will contain much fewer parameters but still achieve good performance (as shown in Figure 4). (III) $E_g$: there is no need to force $E_g$ be orthogonal since an identity matrix is added to the GCN. (IV) Learned graph: the values in the learned adaptive graph vary from 0.0014 to 0.0396 (for PeMSD4), which demonstrates that DAGG can reveal relative importance among different series. We will visualize the learned adaptive graph with heatmap and plot some highly-correlated series in the supplementary. We can achieve sparse graph and avoid negative links by setting small values to zero. However, we didn't see it brings obvious improvements in our experiments. The learned graph in DAGG can perform well and we will keep exploring more advanced graph generation methods.

**To Reviewer 2:** Thanks for your careful review. (I) Experiments: In Table 1, the "*" sign should be presented to the results of STSGCN. As you have listed, DCRNN, STGCN, and ASTGCN used different datasets for evaluation. We followed the most recent works (e.g., ASTGCN and STSGCN) and conducted experiments on the PeMSD4 and PeMSD8 datasets. Our results can be cross-checked and compared with the results presented in ASTGCN and STSGCN. (II) Metr-LA: the Metr-LA dataset is not the first choice for our evaluation because prior works in Metr-LA (e.g., DCRNN) use early fusion (e.g., concatenation) to integrate the extraneous data (e.g., time of day), which is not the focus of our work and does not help the graph generation process. However, we still conduct experiments on Metr-LA following your suggestion and compare our results with the reported results in DCRNN (will add to the supplementary). Experiments show that AGCRN achieves comparable performance to DCRNN and consistently outperforms GCRNN (i.e., DCRNN with undirected graph). The long-term prediction performance of AGCRN is especially superior (for 1 hour ahead prediction, MAE:3.57, RMSE:7.38, MAPE:10.09%). Our further analysis show AGCRN can gain more advantages over DCRNN under a fair comparison (i.e., with the extraneous data removed for both models), even though AGCRN does not rely on encoder-decoder, teacher forcing, or learning rate decay strategies while DCRNN does. (III) Our preliminary experiments show that pair-wise distance or similarity-based graph can work together with DAGG under a multi-graph framework and provide extra information. Also, we agree that generating outputs in a sequential manner (e.g., under the encoder-decoder framework) may improve the results despite sacrificing efficiency. It is convincing that AGCRN has potentials to enable more advanced performance in time series analysis. (IV) Indeed, learning in large graphs for evolving systems is important yet not well studied in literature. Graph partition and sub-graph training may probably work with some adaption. We will highlight this direction as a future work.

**To Reviewer 3:** Thanks for your comments. (I) Loss function: unfortunately, there is no standard about which loss function is better. DCRNN uses L1 loss; STSGCN uses Huber Loss; STGCN uses L2 loss. We chose L1 loss because it achieved better results in our experiments than L2 loss. We would like to highlight that we have carefully chosen the best settings (e.g., loss function) for the baselines to make sure our comparison is fair unless stated otherwise. (II) Prediction Length: first, each step is 5 mins in our experiments (see line 196 of page 5). Thus, we predict the future 60 mins data with the historical 60 mins data, which is the most common setting used for traffic forecasting. Second, our problem is formulated to predict several future steps data with multiple historical steps data (Section 3.1). Thus, the exact time lengths of the historical time period and future period are dependent on the number of time steps and the length of one step, which could vary by applications. (III) Multi-step prediction: Yes, the results in Section 4.4 come directly from the results of multi-step prediction. As formulated in Section 3.1 and mentioned in Section 3.5, our method can generate multi-step predictions directly. With 5 mins as the slot length in our experiments, the short-term (5 mins) and long-term (60 mins) predictions refer to the predictions made at the first step and the $12^{th}$ step, respectively.

**To Reviewer 4:** Thanks for your comments. (I) we omitted some works about GCN for classification tasks due to the space limitation and different problem settings. We will add a subsection under Section 2 to discuss those omitted studies. Specifically, our work deal with dynamically evolving streams from nodes in the whole graph but GCN-based classification models deal with static features/attributes from nodes within a neighborhood. Our work may share some high-level similarities with these studies but has totally different designs for different purposes. For NAPL, our work aim to learn unique parameters for each node, while GAT aims at learning different importance scores for neighbors. The weights in GAT are still shared among different sub-graphs. The matrix factorization in NAPL may be similar with clustering in graph pooling at a high level, but they work in different ways. For DAGG, the learned node embedding in our work can be regarded as high-level abstraction of the whole data stream and is used to generate the graph. But in GCN-based classification models, the learned node embedding is the transformation of nodes feature with the help of existing graph. (II) we apologise for the typos. Metr-La should be PeMSD8 and STSGCN should be ref [11]. We have revised them and conducted a thorough proofreading to eliminate possible typos and grammar mistakes.

[Meta-Review · NeurIPS 2020]

This paper studies the problem of traffic flow forecasting. The proposed ideas are interesting and the evaluation is convincing. The authors are suggested to improve the paper in presentation and justification of the proposed approach in the final version.